# Virtual tissue microstructure reconstruction across species using generative deep learning

**Nicolás Bettancourt**[1,2,3☯], **Cristian Pérez-Gallardo**[1,2], **Valeria Candia**[1,2],
**Pamela Guevara**[3], **Yannis Kalaidzidis**[4], **Marino Zerial**[4], **Fabián Segovia-Miranda**[1,2]*,
**Hernán Morales-Navarrete**[5,6☯]*

1 Faculty of Biological Sciences, Department of Cell Biology, Universidad de Concepción, Concepción, Chile,
2 Faculty of Biological Sciences, Grupo de Procesos en Biología del Desarrollo (GDeP), Universidad de Concepción, Concepción, Chile, 3 Faculty of Engineering, Department of Electrical Engineering, Universidad de Concepción, Concepción, Chile, 4 Max Planck Institute of Molecular Cell Biology and Genetics, Dresden, Germany, 5 Department of Systems Biology of Development, University of Konstanz, Konstanz, Germany, 6 Facultad de Ciencias Técnicas, Universidad Internacional Del Ecuador UIDE, Quito, Ecuador

☯ These authors contributed equally to this work.
* fabiansegovia@udec.cl (FS-M); hernan.morales-navarrete@uni-konstanz.de (HM-N)

**Data Availability Statement:** All the image data used for training and test is available at https://zenodo.org/uploads/10200939.

## Abstract

Analyzing tissue microstructure is essential for understanding complex biological systems in different species. Tissue functions largely depend on their intrinsic tissue architecture. Therefore, studying the three-dimensional (3D) microstructure of tissues, such as the liver, is particularly fascinating due to its conserved essential roles in metabolic processes and detoxification. Here, we present TiMiGNet, a novel deep learning approach for virtual 3D tissue microstructure reconstruction using Generative Adversarial Networks and fluorescence microscopy. TiMiGNet overcomes challenges such as poor antibody penetration and time-intensive procedures by generating accurate, high-resolution predictions of tissue components across large volumes without the need of paired images as input. We applied TiMiGNet to analyze tissue microstructure in mouse and human liver tissue. TiMiGNet shows high performance in predicting structures like bile canaliculi, sinusoids, and Kupffer cell shapes from actin meshwork images. Remarkably, using TiMiGNet we were able to computationally reconstruct tissue structures that cannot be directly imaged due experimental limitations in deep dense tissues, a significant advancement in deep tissue imaging. Our open-source virtual prediction tool facilitates accessible and efficient multi-species tissue microstructure analysis, accommodating researchers with varying expertise levels. Overall, our method represents a powerful approach for studying tissue microstructure, with far-reaching applications in diverse biological contexts and species.

## Introduction

The elucidation of tissue microstructure, encompassing the intricate arrangement and organization of cells, extracellular matrix, and other components within biological tissues, holds paramount significance in biological research [1–4]. Of particular interest is the exploration of

**Funding:** This work was financially supported by Fondo Nacional de Desarrollo Científico y Tecnológico, ANID Fondecyt regular 1200965 to FS-M, VRID-UdeC 2024001079INV to FS-M, and ANID-Basal Center FB210017 (CENIA) to PG. The funders had no role in study design, data collection and analysis, decision to publish, or preparation of the manuscript.

three-dimensional (3D) tissue microstructure, such as that observed in the liver, given its intricate organization and indispensable roles in metabolic, detoxification, and synthesis processes [5,6]. The liver, comprising diverse cell types including hepatocytes, stellate cells, endothelial cells, and immune cells, exhibits a remarkably ordered arrangement [7–9]. The spatial distribution and interactions among these cells play a critical role in governing the liver's proper functionality [10,11]. The hepatocytes together with the sinusoidal endothelial cells form complex tissue structures which optimizes the exchange of nutrients and bile secretion [11].

Unraveling the 3D microstructure of the liver and other tissues offers invaluable insights into tissue development, disease progression, and therapeutic responses [6]. Perturbations in tissue microstructure often manifest during pathological conditions, including fibrosis, inflammation, and tumor growth [12]. In-depth investigations of such alterations facilitate a deeper comprehension of disease mechanisms and the identification of potential targets for diagnostic and therapeutic interventions [13,14]. Yet, capturing and reconstructing the 3D tissue microstructure accurately presents considerable challenges, even when using both traditional and advanced methodologies [15]. Reconstructing digital tissue models requires simultaneous imaging of multiple markers, such as antibodies, chimeric proteins, and small fluorescent molecules, across extensive volumes. The limitations of this endeavor encompass obstacles like inadequate antibody penetration, restrictions on the number of concurrently imaged fluorescent markers, and considerable acquisition time encompassing sample preparation and imaging procedures [15,16]. Addressing these bottlenecks holds the key to advancing the field of 3D tissue modeling.

Over the past decade, deep learning has emerged as a groundbreaking technique in machine learning, showcasing its remarkable ability to unravel intricate patterns and representations from vast and complex datasets [17]. Deep learning has revolutionized the realm of artificial intelligence by eliminating the need for manual feature engineering. It employs artificial neural networks, inspired by the intricate connectivity of biological neural networks in the human brain, to automatically learn and discern intricate features directly from the data [17,18]. By leveraging multiple layers of interconnected nodes, deep neural networks harness their hierarchical structure to progressively extract increasingly abstract and meaningful representations of the input data [19]. This learning technique has achieved unprecedented successes in critical tasks such as image classification, object detection, and image segmentation [20]. A distinctive area where deep learning's prowess shines prominently is in the field of biology, particularly the analysis of microscopy images [18,21,22]. Deep learning models applied to microscopy image analysis have showcased their capability to automatically segment and classify cells, track their dynamic movements, and extract intricate features [18,21–27]. This transformative technology empowers researchers to investigate cellular dynamics, identify disease biomarkers, and expedite the discovery of novel therapeutics. Here, we propose an approach for virtual tissue microstructure reconstruction through the integration of Generative Adversarial Networks (GANs) [28] and fluorescence microscopy. By harnessing the strengths of GANs and leveraging the high-resolution imaging capabilities offered by fluorescence microscopy, we aim to provide a simple yet powerful tool to extract tissue micro-structural information from microscopy images of the actin meshwork. As a proof of principle, we reconstructed detailed and accurate reconstructions of the 3D tissue microstructure of liver samples of different species such as mouse and human. Our proposed approach holds high potential for enabling a deeper understanding of tissue microstructure in healthy and potentially diseased states.

## Materials and methods

### Animals and ethical approval

Adult C57BL/6J mice (8–10 weeks old) were obtained from the animal facility (Centro Regional de Estudios Avanzados para la Vida (CREAV)) at the Universidad de Concepción. The animals were maintained in strict pathogen-free conditions and received *ad libitum* feeding. All procedures performed were approved by the vice rectory of ethics and biosecurity committee from the investigation and development of Universidad de Concepción(permission number: CEBB 635–2020).

### Mice liver collection and immunostaining

Animals were euthanized through transcardial perfusion using PFA. Initially, mice were anesthetized with an intraperitoneal injection of 90 mg/kg body weight ketamine and 10 mg/kg body weight Rompun. Surgical scissors were used to open the abdominal and thoracic cavities, and a 32G needle was inserted into the posterior end of the left ventricle. The needle was secured with a clamp to prevent PFA leakage. A small incision was then made in the right atrium to allow the blood to exit the circulatory system. The animals were perfused at a rate of 3.7 ml/min for 10–15 minutes with a solution of 4% paraformaldehyde, 0.1% Tween-20, and 1x PBS using a peristaltic pump.

### Imaging

Liver samples were imaged (0.3 μm voxel size) in an inverted multiphoton laser-scanning microscope (Zeiss LSM 780) using a 40x1.2 numerical aperture multi-Immersion objective (Zeiss). DAPI was excited at 780 nm using a Chameleon Ti-Sapphire 2-photon laser. Alexa Fluor 488, 555 and 647 were excited with 488, 561 and 633 laser lines and detected with Gallium arsenide phosphide (GaAsp) detectors.

### Image pre-processing

The different components of liver tissues (BC, sinusoids and cortical mesh) were imaged with high-resolution (voxel size 0.3 x 0.3 x 0.3 μm) fluorescent image stacks (80/100μm depth). To cover the entire CV-PV axes, 2x1 tiles were stitched using the image stitching plug-in of Fiji [29]. The 3D images were first denoised using the PURE-LET algorithm [30] with the maximum number of cycles. Then, a background and shading correction was performed using the tool BaSiC [31] along the stack. Finally, all channels were aligned to the actin mesh channels using the function Correct 3D Drift from Fiji.

### Human liver images

Z-stack images of NAFLD human liver samples were obtained from [6].

### TiMiGNet architecture

The model consists of two Generators and two Discriminators. Generator A to B (G-AB) takes an image (real) of class A as input and generates an image (fake) of class B as output. On the other hand, Generator B to A (G-BA) takes an image (real) of class B as input and generates an image (fake) of class A as output. Discriminator A (D-A) classifies images generated by G-BA as real or fake, whilst Discriminator B (D-B) classifies images generated by G-AB as real or fake. The objective is to train both generators and both discriminators. This process will,

eventually, allow the generators to create realistic enough fake images and deceive the discriminators.

*Generator architecture*: The architecture for models trained with patches of size 128x128 pixels: c7s1-64, d128, d256, R256 (x6), u128, u64, c7s1-1. The architecture for models trained with patches of size 256x256 pixels: c7s1-64, d128, d256, R256 (x9), u128, u64, c7s1-1. Where: c7s1-k is a convolution of k filters of size 7x7 and stride 1, followed by an Instance Normalization (IN) layer and a ReLU layer, dk is a convolution of k filters of size 3x3 with stride 2, followed by an IN layer and a ReLU layer, Rk is a residual block with two convolutions with equal number of filters of size 3x3, uk is a block of a transposed convolution with k filters of size 3x3 and stride 2, followed by an IN layer and a ReLU layer. Discriminator architecture: The architecture for both discriminators is: C64 - C128 - C256 - C512 - F1. Where: Ck is a block of a convolution of k filters of size 4x4, followed by an IN layer and a LeakyReLU layer, F1 is a convolution of 1 filter of size 4x4.

The network for the 3D model is essentially the same as the 2D model, with minor differences only: the 2D input and output layers were adapted to accept and produce 3D patches, respectively, and all 2D convolution layers were swapped for 3D convolution layers.

In the case of TiMiGNet+, the major difference relies on how the loss function is defined for the training farmework. Whereas for TiMiGNet, the Mean Squared Error (MSE) over the whole images was used as the loss function, for TiMiGNet+, the background and foreground of the images as split during the training process using the Otsu method. Then the MSE was calculated independently for the background and foreground images, and the loss function was defined as the weighted sum of the MSE. Here we used 0.2 and 0.8 as weights background and foreground, respectively.

## Model training

TiMiGNet *2D and* TiMiGNet+ *2D*: Four images, two for each domain, were used to train the 2D models. Pixel values were normalized between -1 and 1. 2500 two-dimensional patches were extracted from each image, leading to 5000 patches per domain. Patches of size 128x128 pixels were used for the membrane/BC and the membrane/sinusoids models, whereas patches of size 256x256 pixels were used for the membrane/Kupffer model. All models were trained for 100 epochs using Adam optimizer with a learning-rate of 0.002, a beta value of 0.5, a batch-size of 1 and Mean Squared Error as the loss function.

TiMiGNet *3D*: Four images, two for each domain, were used to train the 3D model. Pixel values were normalized between -1 and 1. For the membrane/BC model, 900 three-dimensional patches of size 64x64x64 pixels were extracted from each image, leading to 1800 patches per domain. The ninety percent of the patches were used for training and the remaining ten percent were used for validation. Hyperparameters for the 3D model were set to be equal to those of the 2D model, except for the number of epochs which was set to 300. This allowed us to stop the training early if no improvements were observed.

## Quality evaluation metrics

One image of an independent specimen was used as test image, and was were divided in 25 non-overlapping cubes of dimension 128x128x128. The metrics were estimated independently for each cube. We evaluated the predictive power of the models using an extensive set of well-establish metrics including:

Fechet Inception Distance (FID) [1], which measures the similarity between the generated and ground truth images based on features of the raw images calculated using the inception v3

model. FID is calculated by computing the Fréchet distance between two Gaussians fitted to feature representations of this model.

Mean Squared Error (MSE) [2], which measures the average squared difference between the pixel values of an image generated by a model and the pixel values of the ground truth image.

$$MSE = \frac{1}{n} \sum_{i=1}^{n} (y_i - \hat{y_l})^2$$

Where $\hat{y}$ is the predicted image and $\hat{y}$ is the ground truth image, $n$ is the number of pixel/voxels in the images

Mean Squared Logarithmic Error (MSLE) [2] is similar to MSE but is applied to the natural logarithm of the generated and ground truth images. It is often used in tasks where the error distribution is expected to be logarithmic in nature.

Mean Absolute Error (MAE) [2] calculates the average absolute difference between the pixel values of the generated and ground truth images, making it less sensitive to outliers compared to MSE.

Root Mean Squared Error (RMSE) [2] is the square root of the MSE. It provides a measure of the standard deviation of the errors.

Peak Signal-to-Noise Ratio (PSNR) [3] is an expression for the ratio between the maximum pixel value of the ground truth image to the Mean Squared Error between the generated and ground truth images.

Structural Similarity Index Measure (SSIM) [3] is a metric used to assess the structural similarity between two images based on a perception-based model that takes into account image luminance, contrast, and structure.

Multi-scale Structural Similarity Index Measure (MS-SSIM) [4] is an extension of SSIM that considers multiple scales in the image. It provides a more comprehensive assessment of image quality by taking into account variations at different levels.

Cosine Similarity (COS) [5] is a measure used to determine the similarity between two vectors in a multi-dimensional space. It calculates the cosine of the angle between the vectors.

Correlation Coefficient (CoC) [5] is a statistical measure that quantifies the linear relationship between two sets of data points. For 2D images it can be used to assess how closely related or linearly associated are the pixel values of the generated and ground truth images.

### Large Language Models (LLMs)

ChatGPT 3.5 was used to rephrase some sections of the document and to correct grammatical mistakes. The authors carefully checked for scientific consistency of the generated text.

## Results and discussion

### CNNs and GANs accurately predict several tissue structures in mouse liver tissue

Previous studies [25,26] have shown the great potential of in-silico labeling by using deep learning models to predict fluorescent labels in unlabeled microscopy images. However, they are limited to cell culture systems. Here, we used deep learning models to generate in-silico labeling of 3D complex tissue microstructures. We used two well-established deep learning models, namely convolutional neural networks (CNNs) and Generative Adversarial Networks (GANs); and we compared their predictive power when applied to generate virtual images of various components of liver tissue microstructure based solely on high-resolution images of

the actin mesh. These images were acquired using confocal microscopy. We used two network architectures: modified versions of UNet [32] and CycleGAN [33], termed TiMiPNet (Tissue Microstructure Predictor, S1A Fig) and TiMiGNet (Tissue Microstructure Generator Fig 1A), respectively (refer to the Methods section for details). Additionally, we proposed a variation of TiMiGNet, called TiMiGNet+, which uses the well-established Otsu methods to discriminated the background signal from the foreground signal during the training process. TiMiPNet was trained using a dataset consisting of spatially registered pairs of 3D images of both the input and target structures in mouse liver tissue. Whereas images of the cell border (i.e. cortical actin mesh) served as the input, the output images included the Bile Canaliculi (BC) network or sinusoids (one target structure per trained network). Since, TiMiGNet and TiMiGNet+ (Fig 1A and S1B and S1C Fig) do not require paired images, we computationally mixed the image pairs to simulate unpaired images and have a direct comparison with the models that require paired images such as TiMiPNet.

As shown in Fig 1B and S1–S4 Movies, both TiMiPNet and TiMiGNet+ generated outputs that closely resembled the ground truth images for all the predicted tissue structures, indicating their effectiveness in accurately predicting the different components of liver tissue microstructure from the actin mesh images, even in the absence of paired images (i.e. TiMiGNet+). Next, we quantitatively evaluated the predictions of our models by comparing them with ground truth images acquired experimentally (Fig 1B). Fig 1C shows the Fréchet Inception Distance (FID) [34], a widely used measure that assesses the likeness between the images produced by a generative model and the real images (see Methods for details), for TiMiPNet and TiMiGNet+. We calculated the FID in the context of the entire images and by separately considering the background and foreground regions (S2A Fig). Whereas lower values show good predictive power, high ones indicate larger difference between the prediction and the ground truth. As shown in Fig 1C, both TiMiPNet and TiMiGNet show similar predictive levels for all structures (i.e. BC and Sinusoids). However, TiMiPNet+ (PSNR = 21.37 and SSIM = 0.664) over performs TiMiPNet (PSNR = 20.288 and SSIM = 0.620) when evaluated in the prediction of large structures such as Sinusoids (S3 Fig).

To ensure a more comprehensive evaluation of our predictions, we conducted extra extensive tests specifically targeting the accuracy of predicting the BC, also comparing the results of both models in 2D and 3D. For a rigorous quantitative assessment, we employed various well-known evaluation metrics (See methods, S2 Fig and S1 Table). These metrics enabled us to thoroughly analyze the performance of our predictions. Detailed information regarding these evaluations can be found in the Methods section and S2 Fig. Interestingly, our results showed no substantial differences between the 2D and 3D architectures, as depicted in S2 Fig and S1 Table. However, the computational times for training 3D models are much higher than the ones needed for 2D architectures, i.e. up to 3 times higher for similar results. It is noteworthy that the majority of errors were primarily concentrated near the large veins, as illustrated in S5 Fig, as sudden changes in the morphology of the structures (i.e. BC) could be the source of these errors. Remarkably, TiMiGNet+ consistently exhibited similar performance as other deep learning methods that require pairs of images for training such as TiMiPNet and Pix2Pix [35], as shown in S4A and S4B Fig. This indicates that both approaches are equally effective in predicting the components of liver tissue microstructure. However, it is worth mentioning that the major advantage of the TiMiGNet and TiMiGNet+ approach is its ability to generate accurate predictions without relying on paired images. This characteristic enhances its flexibility and practical applicability in various scenarios.

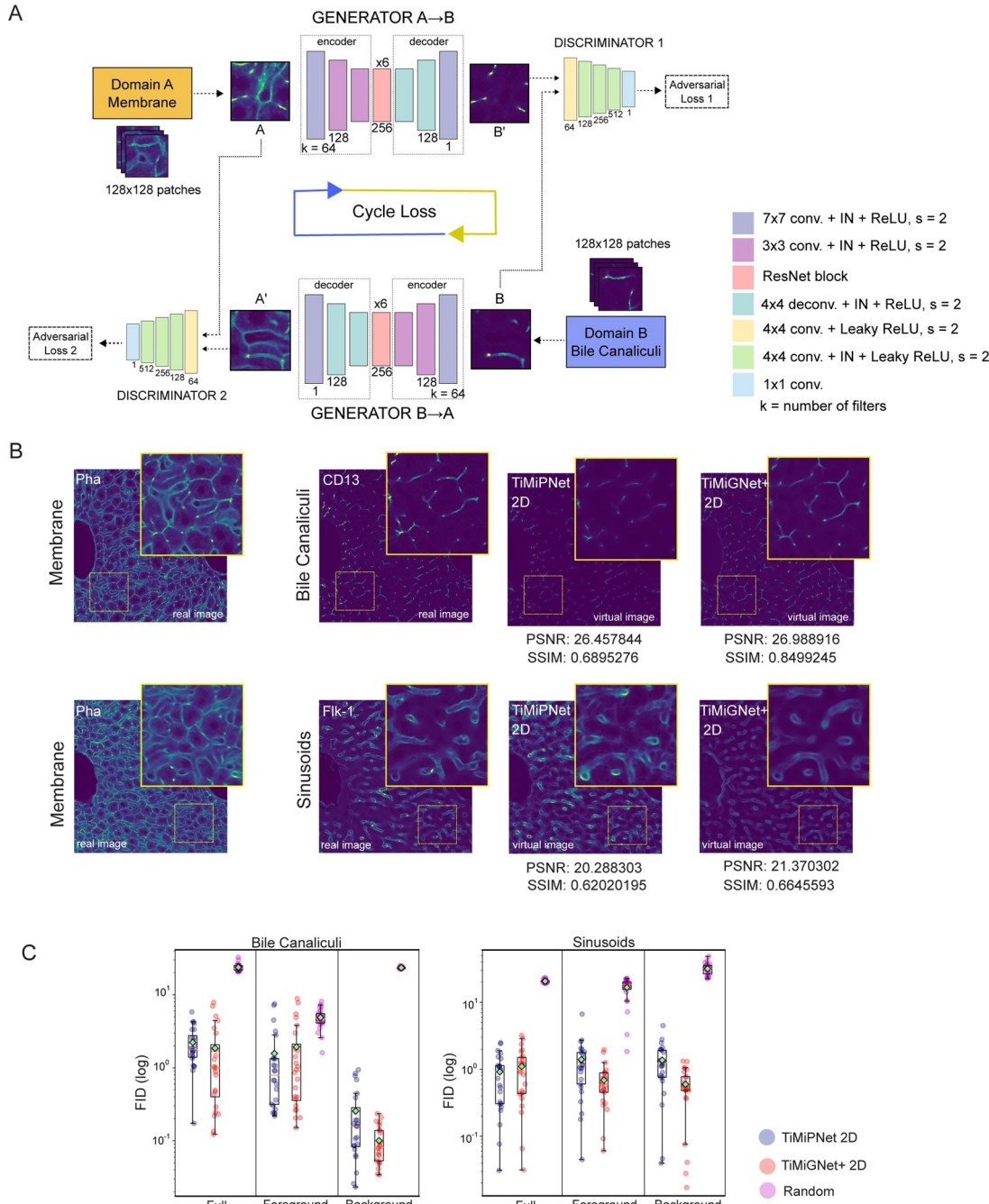

**Fig 1. Deep CNN and GANs accurately predict liver tissue structures from cell border images.** (a) Schematic representation of TiMiGNet. (b) 2D sections of 3D fluorescent images of the actin mesh (membranes), Bile Canaliculi and Sinusoids (experimental images) as well as the corresponding predictions by TiMiPNet and TiMiGNet+ in 2D. c) Quantification of the performance of the predictions of the BC network and sinusoids generated by the models. The test images were divided in 128x128x128 cubes and the metrics were estimated independently for each cube, i.e. each dot represents one image cube. The box plots enclose values from the lower to upper quartiles. The middle line represents the median and the whiskers show the data range (N = 1).

## TiMiGNet accurately predicts BC and sinusoidal networks for deep tissue reconstructions

Given the generality and high performance demonstrated by TiMiGNet in our initial experiments, we proceeded to test its capabilities in more challenging tasks. In particular, we aimed to investigate whether TiMiGNet could effectively predict the intricate structures of the BC and sinusoidal networks in the context of deep tissue imaging. Traditional 3D imaging methods face inherent limitations when it comes to imaging deep tissue regions beyond a depth of approximately 80 to 100 micrometers, especially when utilizing antibody markers for specific structures such as the BC and sinusoids. The limited penetrance of antibodies within the tissue hampers the visualization of these structures beyond the shallow surface layers (Fig 2A, S7–S10 Movies). We took advantage of the unique properties of small fluorescent molecules, such as Phalloidin, which stains the actin mesh of the cells. Phalloidin has the remarkable ability to penetrate several hundred microns into the tissue, surpassing the constraints of antibody staining. We acquired high-resolution images of the actin mesh at a depth of ~240 microns, well beyond the typical imaging range achievable with antibody-based techniques (Fig 2A). Using our pre-trained TiMiPNet and TiMiGNet models, we predicted both the BC and sinusoidal structures from these deep tissue images. Our approach yielded accurate predictions of these structures throughout the entire depth of the images, exceeding the limitations imposed by the restricted antibody staining penetration depth of 80 microns. This is visually depicted in Fig 2A and S7–S10 Movies. Fig 2B and 2C show the mean intensity along the tissue depth for the BC and sinusoids, respectively. Whereas the intensity values for the experimental images, suddenly decreased after ~80 μm, indicating lack of signal; the values for the virtual predictions remained stable along the whole tissue sample, showing both structures (BC and sinusoids) in places where no experimental staining existed (S7–S10 Movies).

The successful application of TiMiGNet in predicting structures beyond the experimentally accessible depth of antibody staining without a need of paired ground truth images represents a great advancement in the field of deep tissue imaging. Our approach not only avoids the need for laborious, expensive and time-consuming antibody-based techniques but also offers a significant reduction in imaging costs. Moreover, by enabling the reconstruction of tissue structures that were previously inaccessible using conventional methods, our approach opens new avenues for exploring the intricate details of deep tissue microarchitecture. The ability to obtain deep tissue reconstructions using a readily available marker, such as Phalloidin, streamlines experimental procedures and expedites data acquisition. Furthermore, the successful prediction of structures beyond the limitations of antibody staining holds great potential for numerous research fields and clinical applications, revolutionizing our understanding of deep tissue biology and paving the way for new discoveries.

## Accurate prediction of tissue microstructure in human liver tissue

The general applicability of small dyes like phalloidin and DAPI enables the staining of tissues across various species, making them invaluable tools, especially when working with immunofluorescence, where challenges often arise due to the limited availability of antibodies that effectively function. To further push the boundaries of our approach, we sought to tackle an even more challenging task: predicting the bile canaliculi structures in human liver tissues. Overcoming this challenge required addressing the practical limitation of not being able to obtain paired images of the actin mesh and BC simultaneously.

For the human samples, we used images sourced from Segovia-Miranda et al. [6]. In their study, the authors encountered a challenge as only one antibody was found to effectively stain the BC network, and it requires an antigen retrieval protocol for optimal performance.

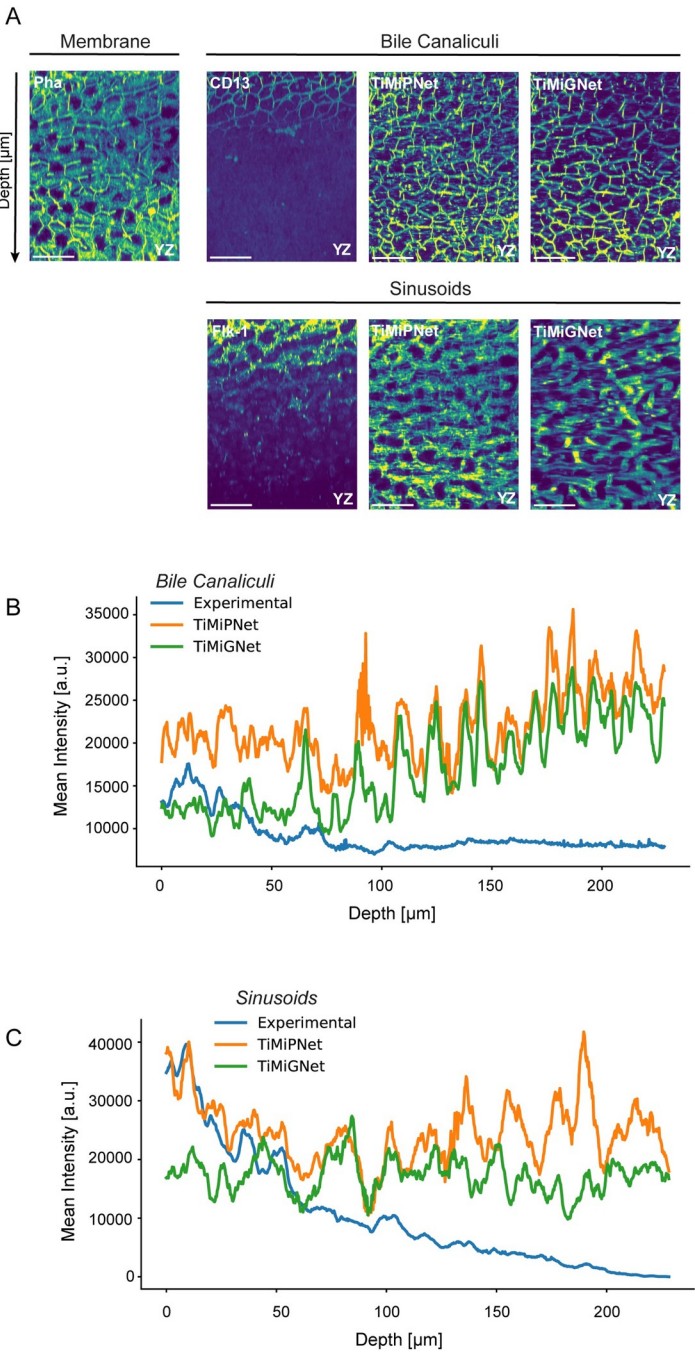

**Fig 2. Deep tissue reconstructions using TiMiGNet.** (a) 2D maximum projections of ~20 µm of the axial sections of 3D fluorescent images of the actin mesh (membranes), Bile Canaliculi, Sinusoids (experimental images) as well as the predictions of TiMiPNet and TiMiGNet. (b) Quantification of the BC signal (mean intensity) along the tissue depth, in the corresponding images of panel a. (c) Quantification of the Sinusoids signal (mean intensity) along the tissue depth, in the corresponding images of panel a (N = 1).

However, this protocol disrupts the actin mesh, rendering phalloidin staining ineffective. Since simultaneous imaging of the actin mesh and BC in human tissue samples was deemed unfeasible, we trained our TiMiGNet model using separate sets of images representing the

actin mesh and BC from different samples and patients. TiMiGNet exhibited a remarkable ability to generate visually convincing predictions of the BC in human liver tissue, as demonstrated in Fig 3B-middle and S11 and S12 Movies. Moreover, we tested if the TiMiGNet trained in mouse tissue samples could also predict BC structures in human tissue. Surprisingly, TiMiGNet (trained in mouse tissue images) was also able to produce highly accurate predictions despite being originally trained on data from a different species (Fig 3B, bottom, and S11 and S12 Movies). We quantitatively evaluate the predictions by comparing tissue parameters such as network radius and branch length estimated from experimental images and virtual predictions of the same patients but different tissue samples. Fig 3B–3D shows remarkably similar values for the predictions, despite the sample-to-sample variability previously shown in [6]. The values of the network radius and branch length were also evaluated along the liver lobule (CV-PV axis) to test spatial variability (Fig 3C and 3D). It is worth noting that, even though the TiMiGNet model trained in mouse samples underestimates the BC radius in ~10/20% , other morphological features such as the branch length are properly predicted. The fact that the network trained on mouse data could successfully generalize its predictions to human tissue underscores the remarkable conservation of BC structures across species. These findings provide valuable insights into the structural similarities and functional significance of BC in both mouse and human liver biology. The successful application of TiMiGNet approach in predicting BC structures in the absence of paired images highlights the adaptability and

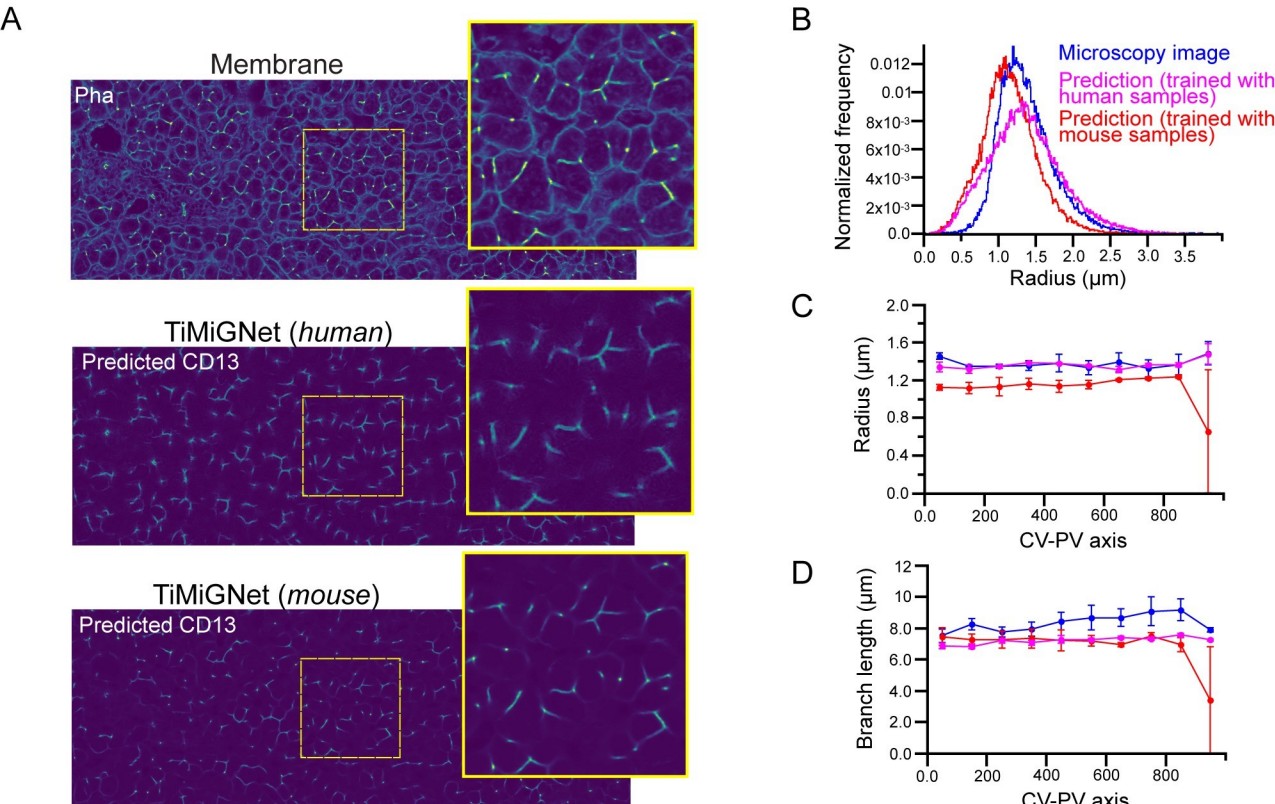

**Fig 3. Prediction of tissue microstructure in human liver tissue using TiMiGNet.** b) 2D sections of 3D fluorescent images of the actin mesh (membranes) and Bile Canaliculi (experimental images) of human liver tissue together with the corresponding predictions of the TiMiGNet 2D model trained in mouse and human tissue images. c-d) Quantification of morphological BC parameters for experimental images and TiMiGNet predictions: Radius distribution(b), mean radius (c) and mean branch length (d) along the CV-PV axis. The error bars represent the standard deviations of the values in the region (N = 1).

robustness of our method. Moreover, it suggests that the underlying architecture and organization of BC are highly conserved across different species.

## Conclusions

Deep-tissue imaging and 3D reconstruction play pivotal roles in advancing our understanding of tissue architecture in both homeostasis and disease conditions. They enable the identification of different tissue characteristics, such as aberrant cell distribution, tissue remodeling, and the formation of disease-specific microenvironments. Current methods often suffer from limitations (poor antibody penetration, restrictions on fluorescent markers), require significant expertise, and can be time-consuming as they typically involve simultaneous imaging of multiple markers across large volumes [16]. Previously, we used CNNs to predict tissue structures by learning features embedded within single-marker images [9,36]. In particular, our deep learning framework showed remarkable accuracy for the prediction of the bile canaliculi (BC) and sinusoidal networks from images of the actin meshwork of liver tissue. However, this approach has several limitations: i) requiring a pair of images of the BC/Sinusoids and the actin mesh, which is not always technically possible, ii) using a 2D approach to predict 3D structures could cause loss of data information, iii) is limited to healthy mouse liver tissue. Here, we overcame these issues and generalized the approach by using Generative Adversarial Network, TiMiGNet. We showed that our proposed methodology has the capability to produce precise and high-resolution predictions of various tissue components, such as bile canaliculus, and sinusoids, based on cell border images (actin meshwork), thereby facilitating efficient and dependable analysis. Moreover, we conducted tests to determine the capability of our networks in predicting more intricate tissue microstructures, specifically the shape of the Kupffer cell, solely based on images of the cell membrane. As illustrated in Fig 4A and S5 and S6 Movies, both TiMiPNet and TiMiGNet produced outputs resembling the ground truth images of Kupffer cell shapes. It is important to note that the predictive performance of the models (Fig 4B and 4C) is not as robust as that observed for other predictions, such as BC and sinusoids. However, these results highlight the method's potential to predict multiple structures from a single-channel image. The integration of TiMiGNet with fluorescence microscopy allowed us to predict tissue structures in scenarios where paired images of ground truth were not attainable. For instance, we demonstrated the potential of using TiMiGNet in predicting structures beyond the experimentally accessible depth of antibody staining, representing a significant advancement in the field of deep tissue imaging. Moreover, we showed that TiMiGNet facilitated multi-species analysis (mouse, human). This makes TiMiGNet a powerful and versatile tool for researchers and practitioners in various domains of biology and medicine.

The utilization of a simple marker, such as Phalloidin, in our method further enhances the practicality and broadens the applicability of our method. Unlike the complex and time-consuming procedures required for simultaneous imaging of multiple markers, our approach relies on a single marker, making it more efficient and cost-effective. This simplicity enables researchers to obtain deep tissue reconstructions with ease, providing a valuable tool for investigating tissue microstructure. Moreover, our method could potentially even be applied in the absence of specific markers, if manually annotated structures are provided as input. This flexibility would allow to utilize our approach in situations where specific markers may not be available or suitable.

The ability to predict BC structures in human liver tissue using TiMiGNet, despite the inherent challenges and limitations, represents a significant advance. This achievement not only could help us to expand our understanding of liver tissue microarchitecture in the context of human biology but also opens up new avenues for investigating the role of BC in various

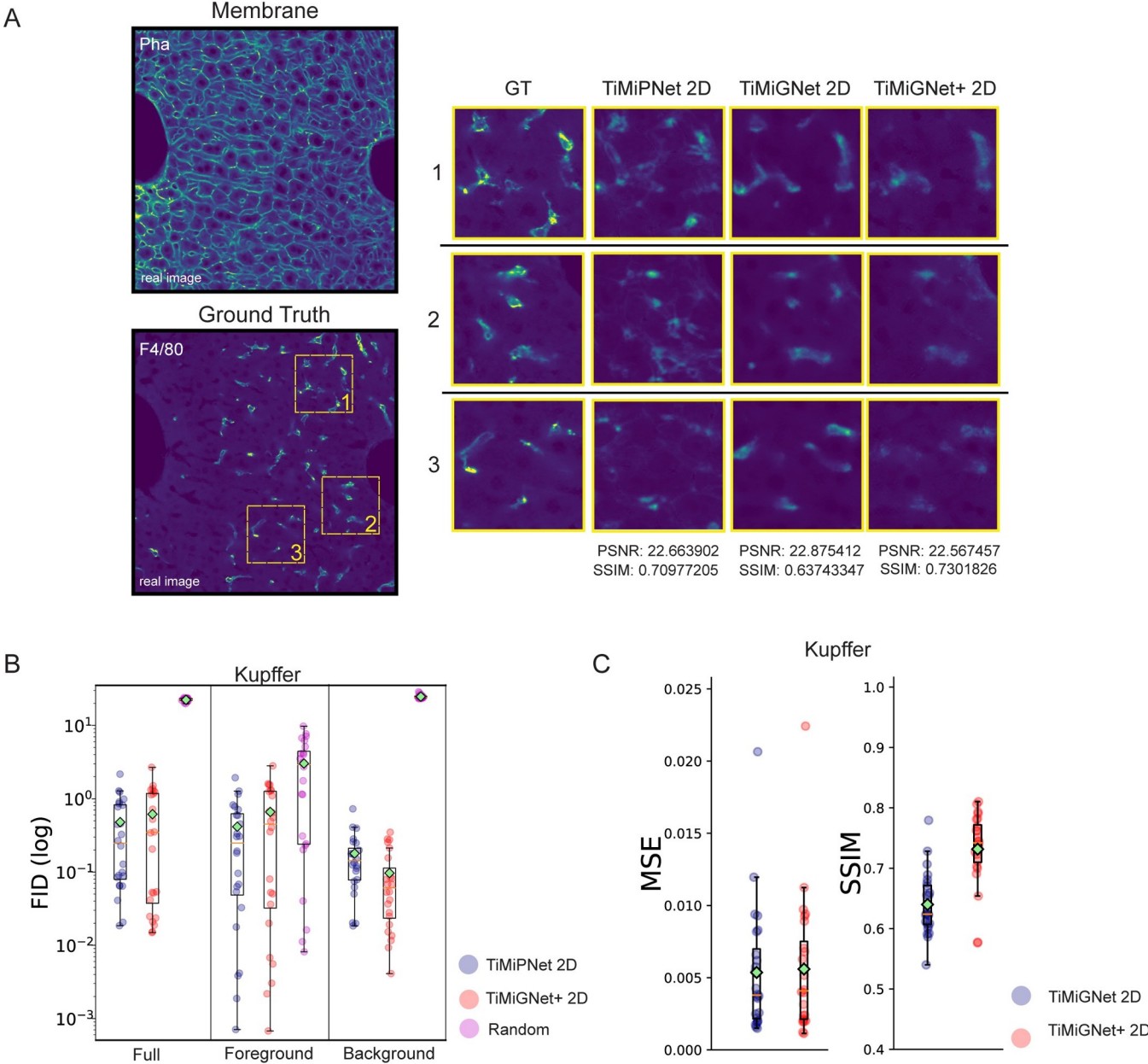

**Fig 4. Prediction of Kupffer cells shapes in liver tissue using TiMiGNet+.** a) 2D sections of 3D fluorescent images of the actin mesh (membranes), Kupffer cells (experimental images) as well as the corresponding predictions by TiMiPNet, TiMiGNet, TiMiGNet+ in 2D. b-c) Quantification of the performance of the predictions of the Kupffer cells generated by the models using FID (b) as well as Mean Squared Error and b) Structural Similarity Index Measure (c). The test images were divided in 128x128x128 cubes and the metrics were estimated independently for each cube, i.e. each dot represents one image cube. The box plots enclose values from the lower to upper quartiles. The middle line represents the median and the whiskers show the data range (N = 1).

liver diseases and clinical applications. Our results demonstrate the power of leveraging computational methods and deep learning techniques to overcome experimental constraints and provide valuable insights into complex biological systems. The successful translation of our approach from mouse to human liver tissue holds great promise for advancing our understanding of liver biology and facilitating the development of novel diagnostic and therapeutic strategies for liver-related disorders.

TiMiGNet could potentially enhances the discovery and analysis of liver diseases by providing detailed virtual reconstructions of tissue microstructures, addressing traditional imaging limitations. It aids in detecting early signs of fibrosis, non-alcoholic fatty liver disease (NAFLD), or liver tumors by accurately predicting and visualizing structures such as bile canaliculi and sinusoids. TiMiGNet's ability to predict bile canaliculi (BC) structures in human liver tissue, despite challenges, marks a significant advancement. This not only deepens our understanding of liver tissue microarchitecture in human biology but also opens avenues for exploring BC's role in various liver diseases and clinical applications. The results demonstrate the power of computational methods and deep learning to overcome experimental limitations, offering valuable insights into complex biological systems. Successfully applying this approach from mouse to human liver tissue promises advancements in liver biology and the development of novel diagnostic and therapeutic strategies.

However, using generative deep learning models in this kind of problems can also have some disadvantages, as previously shown in You et al. [37] for ophthalmology image domains. For instance, collapse might occur, causing the generator to produce a limited variety of samples, reducing the diversity necessary for accurate biological representation. This could result in models that fail to capture the full complexity of tissue structures. Moreover, unintended changes might occur due to differences in data distribution between two domains, leading to artifacts and inconsistencies. These domain shifts are particularly problematic when applying models trained on one species to another, or when the amount of data is reduced like in the case of minor cell populations as Stellate cells. Addressing these potential disadvantages is crucial for enhancing the reliability of deep learning models in tissue microstructure analysis.

Our study presents a novel approach for analyzing tissue microstructure using well-established yet powerful deep learning models and demonstrates its effectiveness in predicting structures beyond the limitations of conventional methods. The simplicity and versatility of our method, particularly with the use of a single marker or minimal annotations, make it a valuable tool for research in various systems. TiMiGNet is made available to the community as open-source software through our GitHub repository (http://github.com/hernanmorales-navarrete/TiMiGNet) and includes the data sets used for training and testing at https://zenodo.org/uploads/10200939. Its straightforward yet efficient design allows for seamless adaptation to diverse applications, ensuring its versatility for various purposes. Future improvements, such as expanding the training dataset, using different network architectures, have the potential to further enhance the accuracy and reliability of our predictions, ultimately advancing our understanding of tissue microarchitecture and its implications in biological systems.

## Supporting information

**S1 Fig. Schematic representation of the used networks.** a) Schematic representation of the training framework for TiMiPNet. b) Schematic representation of the prediction framework for TiMiPNet. c) Schematic representation of the prediction framework for TiMiGNet and TiMiGNet+.
(TIF)

**S2 Fig. Quantification of the performance of the predictions.** a) Schematic representation of the process for generating the mask to split background from foreground in ground truth samples. The masks were calculated using the Otsu method for binary thresholding. b) Quantification of the performance of the predictions of the BC network generated by the 2D and 3D, TiMiPNet, and TiMiGNet models using the following metrics Mean Squared Error, Mean Squared Logarithmic Error, Mean Absolute Error, Root Mean Squared Error, Peak Signal-to-Noise Ratio, Structural Similarity Index Measure, Multi-scale Structural Similarity Index

Measure, Cosine Similarity, and Coefficient of Correlation. The test images were splitted in 128x128x128 cubes and the metrics were estimated independently for each cube, i.e. each dot represents one image cube. The box plots enclose values from the lower to upper quartiles. The middle line represents the median and the whiskers show the data range.
(TIF)

**S3 Fig. Quantification of the performance of TiMiGNet and TiMiGNet+.** Quantification of a) Mean Squared Error and b) Structural Similarity Index Measure for the predictions generated by TiMiGNet and TiMiGNet+ when compared with the experimental images. The test images were splitted in 128x128x128 cubes and the metrics were estimated independently for each cube, i.e. each dot represents one image cube. The box plots enclose values from the lower to upper quartiles. The middle line represents the median and the whiskers show the data range.
(TIF)

**S4 Fig. Quantification of the performance of TiMiGN predictors and generators.** (a) 2D sections of 3D fluorescent images of the actin mesh (membranes) and Bile Canaliculi (experimental images) as well as the corresponding predictions by TiMiPNet, TiMiGNet+ and Pix2Pix in 2D. b) Quantification of the performance of the predictions of the BC network generated by the models. c) 2D sections of 3D fluorescent images of the BC and actin mesh as well as the corresponding predictions of the generators of TiMiPNet and TiMiGNet+. b) Quantification of the performance of the predictions of the BC network generated by the models.
(TIF)

**S5 Fig. Artificial neural networks show errors close to veins in mouse tissue.** (a-b) Representative images actin mesh (membranes), Bile Canaliculi. (c-f) the images were divided into 9x9 blocks and the mean squared error of the predictions of the different models was calculated and shown as a heat map. Whereas low values show good agreement with the ground truth, high vales show potential mispredictions.
(TIF)

**S1 Movie. Z-stack visualization of predicted BC.** Animation of 2D sections along the axial axis of 3D fluorescent images of the actin mesh (membranes) and Bile Canaliculi (experimental images) and the corresponding predictions of TiMiPNet and TiMiGNet.
(MP4)

**S2 Movie. 3D rendering of predicted BC.** 3D rendering of the experimental an predicted images of mouse Bile Canaliculi.
(MP4)

**S3 Movie. Z-stack visualization of predicted sinusoids.** Animation of 2D sections along the axial axis of 3D fluorescent images of the actin mesh (membranes) and Sinusoids (experimental images) and the corresponding predictions of TiMiPNet and TiMiGNet.
(MP4)

**S4 Movie. 3D rendering of predicted sinusoids.** 3D rendering of the experimental a predicted images of mouse Sinusoids.
(MP4)

**S5 Movie. Z-stack visualization of predicted KCs.** Animation of 2D sections along the axial axis of 3D fluorescent images of the actin mesh (membranes) and Kupffer cells (experimental images) and the corresponding predictions of TiMiPNet and TiMiGNet.
(MP4)

**S6 Movie. 3D rendering of predicted KCs.** 3D rendering of the experimental a predicted images of mouse Kupffer cells.
(MP4)

**S7 Movie. Z-stack visualization of TiMiGNet-predicted BC for deep tissue reconstruction.** Animation of 2D sections along the axial axis of 3D fluorescent images of the actin mesh (membranes) and Bile Canaliculi (experimental images) and the corresponding predictions of TiMiPNet and TiMiGNet for deep tissue imaging.
(MP4)

**S8 Movie. 3D rendering of TiMiGNet-predicted BC for deep tissue reconstruction.** 3D rendering of the experimental a predicted images of mouse BC for deep tissue imaging.
(MP4)

**S9 Movie. Z-stack visualization of TiMiGNet-predicted sinusoids for deep tissue reconstruction.** Animation of 2D sections along the axial axis of 3D fluorescent images of the actin mesh (membranes) and Sinusoids (experimental images) and the corresponding predictions of TiMiPNet and TiMiGNet for deep tissue imaging.
(MP4)

**S10 Movie. 3D rendering of TiMiGNet-predicted sinusoids for deep tissue reconstruction.** 3D rendering of the experimental a predicted images of mouse Sinusoids for deep tissue imaging.
(MP4)

**S11 Movie. Z-stack visualization of TiMiGNet-predicted BC for human tissue.** Animation of 2D sections along the axial axis of 3D fluorescent images of the actin mesh (membranes) and BC (experimental images) and the corresponding predictions of TiMiPNet and TiMiGNet for human liver tissue.
(MP4)

**S12 Movie. 3D rendering of TiMiGNet-predicted BC for human tissue.** 3D rendering of the experimental a predicted images of mouse BC for human tissue.
(MP4)

**S1 Table. Evaluation of different network architectures performance.** Average PSNR, SSIM and MSE on the complete testing datasets of the BC.
(DOCX)

## Acknowledgments

We would like to thank the following Facilities from Universidad de Concepción for their support: Centro de Microscopía Avanzada (CMA BIO-BIO) and Centro Regional de Estudios por la Vida (CREAV).

## Author Contributions

**Conceptualization:** Yannis Kalaidzidis, Marino Zerial, Fabián Segovia-Miranda, Hernán Morales-Navarrete.

**Data curation:** Hernán Morales-Navarrete.

**Formal analysis:** Nicolás Bettancourt, Hernán Morales-Navarrete.

**Funding acquisition:** Fabián Segovia-Miranda.

**Investigation:** Nicolás Bettancourt, Cristian Pérez-Gallardo, Valeria Candia, Hernán Morales-Navarrete.

**Methodology:** Nicolás Bettancourt, Fabián Segovia-Miranda, Hernán Morales-Navarrete.

**Project administration:** Valeria Candia, Fabián Segovia-Miranda, Hernán Morales-Navarrete.

**Resources:** Fabián Segovia-Miranda.

**Software:** Nicolás Bettancourt, Hernán Morales-Navarrete.

**Supervision:** Fabián Segovia-Miranda, Hernán Morales-Navarrete.

**Validation:** Nicolás Bettancourt, Hernán Morales-Navarrete.

**Visualization:** Nicolás Bettancourt, Cristian Pérez-Gallardo, Hernán Morales-Navarrete.

**Writing – original draft:** Nicolás Bettancourt, Fabián Segovia-Miranda, Hernán Morales-Navarrete.

**Writing – review & editing:** Pamela Guevara, Fabián Segovia-Miranda, Hernán Morales-Navarrete.

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
