## [Decision Letter · Decision Letter 0]

11 Apr 2024

PONE-D-24-05099Virtual tissue microstructure reconstruction across species using generative deep learningPLOS ONE

Dear Dr. Morales-Navarrete,

Thank you for submitting your manuscript to PLOS ONE. After careful consideration, we feel that it has merit but does not fully meet PLOS ONE’s publication criteria as it currently stands. Therefore, we invite you to submit a revised version of the manuscript that addresses the points raised during the review process.

Particularly, please consider and address the comments from Reviewer #1 and #2. 

We look forward to receiving your revised manuscript.

Kind regards,

Zhentian Wang, Ph.D.

Academic Editor

PLOS ONE

Journal Requirements:

 [Fondecyt (Grant # 1200965)].  

4. Please expand the acronym “Fondecyt” (as indicated in your financial disclosure) so that it states the name of your funders in full.

5. We notice that your supplementary Supplementary Figures 1-3 are included in the manuscript file. Please remove them and upload them with the file type 'Supporting Information'. Please ensure that each Supporting Information file has a legend listed in the manuscript after the references list.

Reviewers' comments:

Reviewer's Responses to Questions

**Comments to the Author**

1. Is the manuscript technically sound, and do the data support the conclusions?

Reviewer #1: Partly

Reviewer #2: Yes

Reviewer #3: Yes

2. Has the statistical analysis been performed appropriately and rigorously? 

Reviewer #1: Yes

Reviewer #2: Yes

Reviewer #3: Yes

3. Have the authors made all data underlying the findings in their manuscript fully available?

Reviewer #1: Yes

Reviewer #2: Yes

Reviewer #3: Yes

4. Is the manuscript presented in an intelligible fashion and written in standard English?

Reviewer #1: Yes

Reviewer #2: Yes

Reviewer #3: Yes

5. Review Comments to the Author

Reviewer #1: This work reported by Nicolás Bettancourt et al. introduced an approach for virtual tissue microstructure reconstruction through the integration of Generative Adversarial Networks (GANs) and fluorescence microscopy. By harnessing the strengths of GANs and leveraging the high-resolution imaging capabilities offered by fluorescence microscopy, they aim to provide a simple yet powerful tool to extract tissue micro-structural information from microscopy images of the actin meshwork. There have been many published studies on this aspect, and the overall quality of this paper is average. I am of the opinions that this manuscript needs to be revised seriously. Some suggestions were included for the improvement of this manuscript.

(1) What is the trainng and testing frameworks of TiMiPNet, TiMiGNet and TiMiGNet+. Figure 1 only have a training

framework of TiMiGNet. What is the input and output of these networks?

(2) Lack of comparative experiments for predictor and generator. Please compare with other deep learning methods.

(3) Please use a table to show the average PSNR and average SSIM on the complete testing dataset.

(4) The pixels of the data picture are too low, and the text in some pictures can't be seen clearly.

(5) Please indicate the number of images used for training and testing.

(6) Please indicate the PSNR and SSIM under each samples in each figure.

(7) In short, the logic of the paper needs to be reorganized. There are some parameters that need to be explained in

detail.

Reviewer #2: TiMiGNet is an algorithm based on domain translation and is a generative artificial intelligence that allows researchers to see microstructures in images more clearly. This is a well-written paper, and various experiments were conducted to confirm the performance of the algorithm.

1. A quantitative evaluation was conducted, but quantitative numbers were not mentioned in the results section. Specific descriptions of these are needed.

2. Is it possible to compare the performance of the proposed model with the default cyclegan?

3. What diseases can be better discovered using the proposed algorithm? The clinical significance needs to be further elucidated.

4. There are disadvantages of GAN. The following paper introduces the shortcomings. "Application of generative adversarial networks (GAN) for ophthalmology image domains: a survey" Were there any experimental results like this? Discuss further the disadvantages.

a) Mode collapse where the generator produces limited varieties of samples. b) Spatial deformity due to small training images without spatial alignment. c) Unintended changes due to the difference of data distribution between two domains. d) Checker-board artifacts in synthetic images.

Reviewer #3: The manuscript is well written and structured in a clear and logic way to present the new TiMiGNet solution. The authors discussed the methods, results, and debates thoroughly and thoughtfully. The conclusions are strongly supported by the comprehensive evidences. I like the idea of using TiMiGNet for biotissue image recognition and segmentation. The novel approach overcomes some of the existing challenges of biotissue image segmentation. I recommend the editor to accept this manuscript for publication.

One minor edit for authors is to replace all graphs with high resolution images.

6. PLOS authors have the option to publish the peer review history of their article (what does this mean?). If published, this will include your full peer review and any attached files.

Reviewer #1: No

Reviewer #2: No

Reviewer #3: No

---

## [Author Response · Author response to Decision Letter 0]

28 May 2024

We sincerely appreciate the thoughtful and detailed feedback on our manuscript from you and the reviewers. We have carefully considered each of the comments and have made the necessary revisions to address them. Below is a point-by-point response to each of the reviewers' comments:

Reviewer #1:

1. What is the training and testing framework of TiMiPNet, TiMiGNet, and TiMiGNet+? Figure 1 only shows a training framework of TiMiGNet. What is the input and output of these networks?

Thank you for pointing this out. We have added a new figure (S1_Fig A) illustrating the training framework for TiMiPNet. Additionally, we have provided explanations on the differences in the training frameworks of TiMiGNet and TiMiGNet+ (lines 188 to 195). The input for the networks are 64x64 or 128x128 image patches of membranes, and the output for the networks are 64x64 or 128x128 image patches of bile canaliculi. Corresponding figures (S1_Fig B, C) showing the testing/prediction frameworks have also been included.

2. Lack of comparative experiments for predictor and generator. Please compare with other deep learning methods.

We agree that a comparison of several deep learning methods is needed and have included them. Specifically, we used Unet (TiMiPNet), CycleGAN (TiMiGNet), and added Pix2Pix to our analysis. Figures F4 Sup a,b demonstrate similar performance across architectures for predicting BC structures. We have expanded our specific case with CycleGAN by modifying the loss function (TiMiGNet+), achieving better results for large structures like sinusoids (F3 Sup). We also evaluated the generator performance for CycleGAN-like architectures as suggested (F4 Sup c,d).

3. Please use a table to show the average PSNR and average SSIM on the complete testing dataset.

We have added a table displaying the average PSNR and SSIM for the complete testing dataset.

4. The pixels of the data picture are too low, and the text in some pictures can't be seen clearly.

We have exported all images in higher quality and increased the text size for clarity.

5. Please indicate the number of images used for training and testing.

We apologize for the lack of clarity. The number of images and patches used for training and testing are now described in the "Model Training" and "Quality Evaluation Metrics" sections of the Methods.

6. Please indicate the PSNR and SSIM under each sample in each figure.

We have added the average PSNR and SSIM values for each predicted image.

7. In short, the logic of the paper needs to be reorganized. There are some parameters that need to be explained in detail.

Thank you for the comment. We have reorganized the text for better logical flow and provided detailed explanations for relevant parameters.

Reviewer #2:

1. A quantitative evaluation was conducted, but quantitative numbers were not mentioned in the results section. Specific descriptions of these are needed.

Thank you for the comment. We clarified this by adding a summary table (S1 Table) of the average algorithm performance results and incorporated specific quantitative numbers into the results section.

2. Is it possible to compare the performance of the proposed model with the default CycleGAN?

We performed a detailed comparison as explained in our response to Reviewer #1, point 2.

3. What diseases can be better discovered using the proposed algorithm? The clinical significance needs to be further elucidated.

Thank you for raising this point. We have added a paragraph discussing the clinical significance and the potential diseases that can be better discovered using the proposed algorithm (lines 496 to 508).

4. There are disadvantages of GAN. The following paper introduces the shortcomings. "Application of generative adversarial networks (GAN) for ophthalmology image domains: a survey" Were there any experimental results like this? Discuss further the disadvantages.

Thank you for raising this point. We have addressed the disadvantages of GANs, as discussed in "Application of Generative Adversarial Networks (GAN) for Ophthalmology Image Domains: A Survey". We included a discussion on mode collapse, spatial deformity, unintended changes due to domain differences, and checkerboard artifacts in the final section of the manuscript (lines 510 to 521).

Reviewer #3:

The manuscript is well written and structured in a clear and logical way to present the new TiMiGNet solution. The authors discussed the methods, results, and debates thoroughly and thoughtfully. The conclusions are strongly supported by the comprehensive evidence. I like the idea of using TiMiGNet for biotissue image recognition and segmentation. The novel approach overcomes some of the existing challenges of biotissue image segmentation. I recommend the editor to accept this manuscript for publication.

One minor edit for authors is to replace all graphs with high-resolution images.

We appreciate your positive feedback and recommendation for publication. We have replaced all graphs with high-resolution images as suggested.

All figures were uploaded in high resolution, but during the automatic document compilation, their quality decreased.

We hope these revisions adequately address all your comments and improve the quality and clarity of our manuscript. Thank you for your valuable feedback and consideration.

Sincerely yours,

Hernán Andrés Morales-Navarrete

---

## [Decision Letter · Decision Letter 1]

12 Jun 2024

Virtual tissue microstructure reconstruction across species using generative deep learning

PONE-D-24-05099R1

Dear Dr. Morales-Navarrete,

We’re pleased to inform you that your manuscript has been judged scientifically suitable for publication and will be formally accepted for publication once it meets all outstanding technical requirements.

Kind regards,

Zhentian Wang, Ph.D.

Academic Editor

PLOS ONE

Additional Editor Comments (optional):

Reviewers' comments:

Reviewer's Responses to Questions

**Comments to the Author**

1. If the authors have adequately addressed your comments raised in a previous round of review and you feel that this manuscript is now acceptable for publication, you may indicate that here to bypass the “Comments to the Author” section, enter your conflict of interest statement in the “Confidential to Editor” section, and submit your "Accept" recommendation.

Reviewer #1: All comments have been addressed

Reviewer #2: All comments have been addressed

2. Is the manuscript technically sound, and do the data support the conclusions?

Reviewer #1: Yes

Reviewer #2: Yes

3. Has the statistical analysis been performed appropriately and rigorously? 

Reviewer #1: N/A

Reviewer #2: Yes

4. Have the authors made all data underlying the findings in their manuscript fully available?

Reviewer #1: Yes

Reviewer #2: Yes

5. Is the manuscript presented in an intelligible fashion and written in standard English?

Reviewer #1: Yes

Reviewer #2: Yes

6. Review Comments to the Author

Reviewer #1: The picture pixels are fuzzy, so it is suggested that the quality of picture data should be improved.

Reviewer #2: The authors revised the paper well.

The authors stated that they used LLM. Should this be indicated in the paper? I am negative about this. Since chatGPT was not used in the research, it is considered unnecessary.

7. PLOS authors have the option to publish the peer review history of their article (what does this mean?). If published, this will include your full peer review and any attached files.

Reviewer #1: No

Reviewer #2: No

---

## [Editor Report · Acceptance letter]

3 Jul 2024

PONE-D-24-05099R1 

PLOS ONE

Dear Dr. Morales-Navarrete, 

I'm pleased to inform you that your manuscript has been deemed suitable for publication in PLOS ONE. Congratulations! Your manuscript is now being handed over to our production team.

Kind regards, 

on behalf of

Prof. Zhentian Wang 

Academic Editor

PLOS ONE